# Walking on Water—A Natural Experiment of a Population Health Intervention to Promote Physical Activity after the Winter Holidays

**DOI:** 10.3390/ijerph16193627

**Published:** 2019-09-27

**Authors:** Jonathan McGavock, Nicole Brunton, Nika Klaprat, Anders Swanson, Dave Pancoe, Ed Manley, Ashini Weerasinghe, Gillian L. Booth, Kelly Russell, Laura Rosella, Erin Hobin

**Affiliations:** 1Department of Pediatrics and Child Health, Max Rady Faculty of Health Sciences, University of Manitoba, Winnipeg, MB R3E 3P4, Canada; 2Diabetes Research Envisioned and Accomplished in Manitoba (DREAM) Research Theme at the Children’s Hospital Research Institute of Manitoba, Winnipeg, MB R3E 3P4, Canada; 3Children’s Hospital Research Institute of Manitoba, Winnipeg, MB R3E 3P4, Canada; 4Winnipeg Trails Association, Winnipeg, MB R3B 0Y6, Canada; 5The Forks, Winnipeg, MB R3C 4L9, Canada; 6The Bartlett Centre for Advanced Spatial Analysis, Faculty of the Built Environment, University College London, London WC1E 6BT, UK; 7Public Health Ontario, 480 University Avenue, Toronto, ON M5G 1V2, Canada; 8MAP Centre for Urban Health Solutions, St. Michael’s Hospital, Toronto, ON M5B 1W8, Canada; 9Dalla Lana School of Public Health, University of Toronto, Toronto, ON M5T 3M7, Canada

**Keywords:** sports and exercise medicine, obesity, epidemiology, public health

## Abstract

*Background*: Very few experimental studies exist describing the effect of changes to the built environment and opportunities for physical activity (PA). We examined the impact of an urban trail created on a frozen waterway on visitor counts and PA levels. *Methods*: We studied a natural experiment in Winnipeg, Manitoba, Canada that included 374,204 and 237,362 trail users during the 2017/2018 and 2018/2019 winter seasons. The intervention was a 10 km frozen waterway trail lasting 8–10 weeks. The comparator conditions were the time periods immediately before and after the intervention when ~10 kms of land-based trails were accessible to the public. A convenience sample of 466 participants provided directly measured PA while on the frozen waterway. *Results*: Most trail users were 35 years or older (73%), Caucasian (77%), and had an annual household income >$50,000 (61%). Mean daily trail network visits increased ~four-fold when the frozen waterway was open (median and interquartile range (IQR) = 710 (239–1839) vs. 2897 (1360–5583) visits/day, *p* < 0.001), compared with when it was closed. Users achieved medians of 3852 steps (IQR: 2574–5496 steps) and 23 min (IQR: 13–37 min) of moderate to vigorous intensity PA (MVPA) per visit, while 37% of users achieved ≥30 min of MVPA. *Conclusion*: A winter-specific urban trail network on a frozen waterway substantially increased visits to an existing urban trail network and was associated with a meaningful dose of MVPA. Walking on water could nudge populations living in cold climates towards more activity during winter months.

## 1. Introduction

In westernized countries, including Canada, the winter holiday season is associated with significant weight gain [1,2] and reduced physical activity [3,4]. The impact of seasonality on weight gain and physical inactivity is amplified in regions with sub-zero temperatures during the winter [2]. Public health strategies that facilitate behaviour modification to counter weight gain and inactivity during the winter months could thus have significant public health implications.

Behavioural interventions delivered during the winter holiday season prevent the 0.5 to 1.0 kg of weight typically gained during this season [5,6]. While efficacious, this approach may not effectively reach the large segments of the population who gain weight during the winter holiday season [2]. The start of the calendar year is widely recognized as a critical time window to support the adoption of healthy behaviours [7,8]. Strategies that can capitalize on this motivation and encourage healthy behaviours could potentially counter the consequences of unhealthy behaviours over the winter holidays [7,9]. Urban centres that experience multiple weeks with temperatures below freezing at the start of the calendar year provide a unique opportunity for such an approach. Transient frozen waterway trails, groomed to support recreational physical activities, are increasingly being created in northern urban centres [10,11,12]. To the best of our knowledge, there are no empirical studies of these frozen waterway trails on healthy behaviours during winter months.

The purpose of this study was first to assess the impact of a frozen waterway trail on user visits and second to estimate physical activity patterns associated with the trail. The main hypothesis was that daily visits to an urban trail network would increase significantly with the creation of frozen waterway trail compared with days without the frozen waterway trail. We also aimed to describe trail user physical activity levels, demographic profiles, and perceived benefits of the trail.

## 2. Design and Methods

### 2.1. Hypothesis and Research Questions:

The main hypothesis for the study was that daily visits to an urban trail network would increase significantly with the creation of frozen waterway trail compared with days without the frozen waterway trail. We also attempted to answer the following research questions: (1) What are the demographic profiles of trail users? (2) How much physical activity are trail users achieving during a regular visit? (3) What are the perceived benefits of using the trail?

### 2.2. Study Design

To test the main study hypothesis, we capitalized on a natural experiment consisting of a seasonal trail built on a frozen waterway in an urban centre during the 2017/2018 and 2018/2019 winter seasons. In addition to the natural experiment, we conducted field surveys of trail users to address the three research questions presented above. We followed TREND reporting guidelines for this quasi-experimental study [13]. The study methods were approved by the Biomedical Research Ethics Board at the University of Manitoba (H2017:232) in accordance with the Declaration of Helsinki. All participants provided written and informed consent prior to participating in field surveys and directly measured physical activity.

### 2.3. Timing of Data Collection

For the main hypothesis, trail user counts were collected for 20–30 days prior to opening the seasonal winter trail network, during the entire intervention period, and for 30–40 days following closure of the seasonal winter trail network, during the 2017/2018 and 2018/2019 winter seasons. Research assistants also collected field data from users on the frozen waterway from 1 February to 10 March 2019.

### 2.4. Population

The population of interest was visitors to the urban trail network. For the main hypothesis, there were no inclusion or exclusion criteria. For the secondary research questions, a convenience sample of adults over the age of 18 who were willing to wear a waist-mounted pedometer and complete a survey during their visit to the trail network were invited to participate.

### 2.5. Intervention Trail Network

The intervention was a seasonal trail network created on two frozen rivers, the Red and Assiniboine Rivers, that lasted ~10 weeks following the winter holiday season (Figure 1A–C). The trail is created and maintained by the local not-for-profit organization that manages the historic site and urban trail network. In each of the two study years, the trail was open to the public at no cost, 24 h per day, within two weeks following Christmas Day and ended approximately 10 weeks later. During the spring, summer, and fall, the waterway is not walkable (Figure 1D–F). Art displays, kilometer markers, and warming huts were provided along the trail as an incentive to use the entirety of the trail. In addition to the main entry point, several entry points were created along the 10 km frozen waterway trail to provide public access to adjacent neighborhoods (purple dots, Figure 2).

### 2.6. Control Trail Network

The frozen waterway trail (purple line, Figure 2) was added to an existing permanent urban trail network consisting of ~10 km of gravel and paved trails that are freely accessible to the public, 24 h a day, 7 days a week, all year around, and at no cost (green lines, Figure 2). The intersection of the two rivers, known as the Forks, marks a historic landmark that has been a meeting place for the Indigenous Peoples (Anishinaabe, Néhiyaw, and Dakota peoples) in the region for six millennia [14]. The Forks and Parks Canada manage and maintain the trail networks along the rivers and throughout the park for 12 months of the year. The control time period of 20–30 days prior to and immediately following the intervention was selected to match, as closely as possible, the weather conditions during the intervention.

### 2.7. Outcomes of Interest

The primary outcome of interest was daily counts of individuals using the trail network during intervention and control periods. User counts were collected objectively using a PYRO-Box people counter (Eco-Counter, Montreal, QC, Canada) throughout the duration of the intervention and control periods. The PYRO-Box counts people by detecting their body heat using passive-infrared pyroelectric technology, and a high-precision lens, and is used frequently to quantify visitors to urban trials [15,16]. It was chosen for its high level of accuracy, data storage capabilities, wireless connectivity, and long battery life [17,18]. The intervention and control trails have multiple access points (Figure 2). The PYRO-Box was placed at the main entry point for both the frozen waterway trail and the control trail network (red dot, Figure 2). Participants were unaware of the PYRO-box throughout the study.

The outcomes for the secondary research questions included the amount of physical activity users achieved while on the frozen waterway, and the perceived impact of the frozen waterway trail on their health. Waist-mounted pedometers (Piezo^®^RD. StepsCount Inc, Deep River Ontario, Canada) recorded total steps and minutes of moderate to vigorous intensity activity during trail visits from individual users [19,20]. The PiezoRD detects steps with a uniaxial piezoelectric sensor at the waist. These pedometers are valid and reliable for an adult population [21,22]. They have not been validated for capturing skating strides. When weather permitted, trail users completed a brief survey regarding their demographic information, the frequency and duration of trail use, and the perceived impact of the trail on their mental and physical health. Trail counts, objective measurements of physical activity, and surveys of users are the most common methods used to assess physical activity-based population health interventions [15,16,17,18,19,20]. When used in concert, data can be triangulated to answer questions relevant to policy makers, including “How often is the intervention being used?”; “What dose/amount of activity are users achieving?”; and “What are users saying about the intervention?”

### 2.8. Additional Variables Collected

Weather data during the intervention and control periods were obtained from publicly available data provided by Environment Canada. Days were categorized as weekday (Monday–Friday) and weekend day (Saturday–Sunday). National or provincial holidays that occurred on week days were classified as a weekend day. Trail users also provided their postal code of residence for geomapping. Data for ethnicity and household income were provided from the 2006 Statistics Canada Census made available by the City of Winnipeg. We employed the dot density mapping technique to reflect the spatial distribution of population density and demographic variation at the neighbourhood scale.

### 2.9. Public Involvement

Members of the public were involved at several stages of the study, including study design, data collection, and data interpretation, through an organization (Winnipeg Trails Association) dedicated to supporting urban trails for physical activity. The organization, represented by its executive director, acted as a co-applicant on the grant and a collaborator throughout the project.

### 2.10. Analyses

Individual-level data were used for all analyses. T-tests were used to compare differences in physical activity outcomes (i.e., steps/visit and minutes/visit) between trail users and days of the week. Multivariate regression models were used to quantify the influence of the intervention, season, and day of the week on trail user counts. Autoregressive integrated moving average models (ARIMA) were fit separately for both years to determine a difference in daily trail counts between the control and intervention periods, and adjusted for mean temperature and day of the week. The autocorrelation function and partial autocorrelation function plots were used to identify the autoregressive and moving average components and the stationarity of the model. Goodness-of-fit tests of the Akaike information criterion (AIC) and Bayesian information criterion were used to select the optimal model and the autocorrelation of residuals was diagnosed using the Ljung–Box test [23]. These tests determined if autocorrelation within the time series was random or biased in some way. An alpha of 0.05 was used as a threshold for significance. All analyses were performed using SPSS (version 25, IBM, Arnmonk, NY, USA) and R package 3.2 (R Foundation for Statistical Computing, Vienna, Austria).

## 3. Results

Images and a map of the frozen waterway trail are provided in Figure 1 and Figure 2. Details for the intervention and control periods are provided in Table 1. The intervention lasted 56 days with a mean temperature of −13 °C in 2017/2018 and 67 days with a mean temperature of −17 °C in 2018/2019.

Demographics of trail users are presented in Table 2. Among 466 trail users who provided directly measured physical activity data, 290 completed surveys. The remaining 176 users were unable to complete surveys as tablets and ball-point pens were frozen during data collection. Among those completing the survey, 58% were female, the majority (60%) reported an annual household income >$50,000, and most (58%) were over 35 years old. A map of trail users’ household neighbourhoods is provided in Figure 3. Surveyed users tended to live in primarily Caucasian and higher income neighbourhoods, reflecting users’ self-reported demographic attributes (Table 2).

### 3.1. Impact of Intervention on Visitors to the Trail Network

Differences in daily user counts between the intervention and control periods are presented in Figure 4. For the 2017/2018 interrupted time series AutoRegressive Integrated Moving Average (ARIMA) model, the AIC was 1852.651 and the Ljung–Box statistic was not significant (Chi-square = 1.37, *p* = 0.9276). For the 2018/2019 interrupted time series ARIMA model, the AIC was 2233.227 and the Ljung–Box statistic was not significant (Chi-square = 9.22, *p* = 0.1005). Collectively, these data reveal minimal autocorrelation and no bias in the residuals in each of the two intervention years. Daily trail network counts increased ~2- to 4-fold during the intervention period in both years (median and IQR: 2449 (1327 to 4128) to 4516 (2364 to 6251) counts/day in 2017/2018 and from 405 (223 to 663) to 1813 (908 to 3397) counts/day in 2018/2019) and returned to pre-intervention control levels immediately following closure of the winter trail network, in both years (1040 (657 to 1952) counts/day in 2017/2018 and 463 (309 to 883) counts/day in 2018/2019). Counts in the 2018/2019 winter season were significantly lower than in the 2017/2018 season (266,581 vs. 182,298, *p* < 0.01). This was likely because of a colder winter season (mean daily temperature −17 °C vs. −13 °C) and a prolonged climactic event known as a Polar Vortex during the months of January and February.

The increase in trail visits was evident on both weekdays and weekend days in each year (Figure 5). An interrupted time series analysis revealed that user counts increased significantly (*p* < 0.001) when the frozen waterway was open (i.e., during the intervention period) and decreased following closure (*p* < 0.001) after adjusting for weekend days and temperature.

### 3.2. Physical Activity Levels

Physical activity (PA) data stratified by gender and day of the week are presented in Figure 6. No differences in PA were observed between males and females during their trail visit; therefore, data were pooled for both sexes. Overall, users accumulated 4195 ± 2205 steps during 39 ± 20 min of activity. Users accumulated 27 ± 18 min of moderate to vigorous intensity PA (MVPA) during their visit. Thirty-seven percent of trail users achieved the daily target of 30 min of MVPA while using the trail network. Visitors during weekdays achieved significantly (*p* < 0.001) more total steps (4796 vs. 3987), minutes of activity (44 vs. 40 min), and MVPA minutes (32 vs. 25 min) compared with visitors on weekend days (Figure 6D–F). Users who were skating achieved similar total activity steps (4150 ± 2231 vs. 4283 ± 2160 steps, *p* = 0.52) and time (40 ± 21 vs. 38 ± 18 min, *p* = 0.38), but significantly less MVPA (25 ± 17 vs. 31 ± 18 min, *p* < 0.001) during their visit.

### 3.3. User Profiles and Perceived Health Outcomes

Most trail users (61%) travelled 15 min or less to use the trail (Table 3). The vast majority of trail users reported using the trail to support recreational physical activity or exercise. Most users reported using the trail network one time per week with the remainder reporting using >2 times per week. Over 90% of trail network users reported using the trail network for more than 30 min per use. Among those completing surveys, 53% believed their health improved marginally or significantly, and 58% reported that their mental/emotional health improved marginally or significantly since using the trail.

## 4. Discussion

This is the first prospective evaluation of a frozen waterway trail on user visits, physical activity levels, and perceived benefits. We found that the creation of an urban trail along a frozen waterway immediately following the winter holiday season attracted ~200,000–250,000 visitors. Trail users achieved a meaningful dose of physical activity, with over one-third achieving the minimum daily threshold to achieve health benefits. These data provide preliminary evidence to support walking on water, by way of a frozen waterway trail network, as a potentially attractive population health intervention to support daily physical activity during the winter.

The use of frozen waterways to support recreational physical activity during winter was first recorded in the early 1500s [24]. The frozen waterway trail studied here attracted over 200,000 visitors each year, representing 2000–4500 visitors daily. This daily visitor rate was 3- to 5-fold higher than any other published evaluation of an urban trail using similar methods [4,16,25,26]. Furthermore, the 3- to 5-fold increase in user counts during the intervention greatly exceeds the effect size observed in previous natural experiments or urban trail expansions [16,25,26,27]. The data presented here suggest that creating a trail along a frozen waterway is a unique population health strategy that attracts large segments of an urban population to be physically active in the winter. Impressively, this effect was evident during the winter season in a northern climate when temperatures were well below freezing.

Population health interventions are theoretically designed to be equitable, but can exacerbate inequities [28]. The “inverse care law” [29] and systematic reviews reveal that intervention-generated inequalities often favour affluent or privileged segments of a population [28], including those designed to increase physical activity [30]. We found that very few users of the frozen waterway trail came from low income homes or were visible minorities (Figure 3). The reason for these differences is unclear, but may relate to difficulties accessing the trail network with public transit or owing to a lack of adequate clothing needed to participate in physical activity during cold days. It may also be that the elements of the frozen waterway were not designed in a way that made it attractive for these segments of the population. As these individuals often experience great inequities on chronic disease [31,32], they would, in theory, benefit the most from access to a trail network that supported daily physical activity. Future interventions should increase awareness, access, and perceived safety of urban trails for marginalized populations to maximize the benefits, particularly for winter-specific urban trails.

The average amount of physical activity achieved by users of the frozen waterway was sufficient to elicit health benefits. The mean number of steps (~4000) and MVPA (~23 min) achieved during each visit, is associated with improved cardiovascular health and possibly weight loss if achieved daily. Weight gain over the winter holiday season is recognized as an important determinant of obesity risk within high-income countries [1,2]. Over a relatively short time frame, individuals gain between 0.5 and 1.0 kg, accounting for 50% to 80% of annual weight gain [33]. Individualized interventions delivered during the holiday season are effective at preventing weight gain [5,6], although they may not be scalable to entire populations. To our knowledge, there are no published reports of population-based approaches to weight maintenance following the winter holiday season. Using data we collected in the field, the estimated energy expenditure on the frozen waterway is 27.2 and 23.2 kcals/6 min for men and women, respectively [34]. Assuming the average observed visit duration of 40 min, reversing excess weight gain of 0.4 to 1.0 kg could potentially be achieved with 17 to 37 visits for men, and 20 to 43 visits for women. This could be achieved within 5–8 weeks if one visited 3–5 times per week. These are crude estimates based on published data from a controlled laboratory setting, and thus may not extrapolate to a real world setting across a more variable population-based sample of individuals, like the one studied here. Controlled experiments are needed to determine the effectiveness of population-level physical activity interventions to counter weight gain following the holiday season.

It is well established that being outdoors, particularly when connected to nature, reduces blood pressure [35] and improves mental wellness [36,37]. An attractive aspect of the present intervention compared with more conventional approaches to increasing activity during the winter season was the increased access to nature. The use of natural frozen waterways may have elicited additional health benefits owing to the increased exposure to nature that were not captured here. More studies are needed to examine the health benefits of outdoor or nature-based interventions during winter months in colder climates.

The population health intervention studied here is strengthened by the use of objective measures and an interrupted time series design. The results are strengthened by the use of objective measures for main outcomes and the fact that results were replicated over a two-year time frame. Despite these strengths, several limitations need to be addressed. First and foremost, there is a risk for unmeasured confounding that could have influenced trail use, which we did not capture. We have attempted to overcome that limitation with the use of an interrupted time series analysis that included very specific pre- and post-intervention control periods that were as similar to the intervention as possible. We also replicated our findings over two consecutive years. Second, data from trail users are at risk of response and participation bias. It is likely that participants that agreed to wear a pedometer are not reflective of the typical trail user and may have been influenced to respond favorably to questions regarding the perceived benefits of the trail. Finally, physical activity data captured while users were skating may be underestimated as the pedometer may not capture all steps/strides as it would with walking. A controlled experiment is needed to confirm these findings.

## 5. Conclusions

The creation of a trail on a frozen waterway led to a significant increase in visitors to an urban trail network. The dose of activity users achieved while on the frozen waterway was within the range required to achieve health benefits. Trail users reported significant health benefits associated with trail use. Frozen waterways are a novel population health intervention to support increased physical activity following the winter holiday season.

## Figures and Tables

**Figure 1 ijerph-16-03627-f001:**
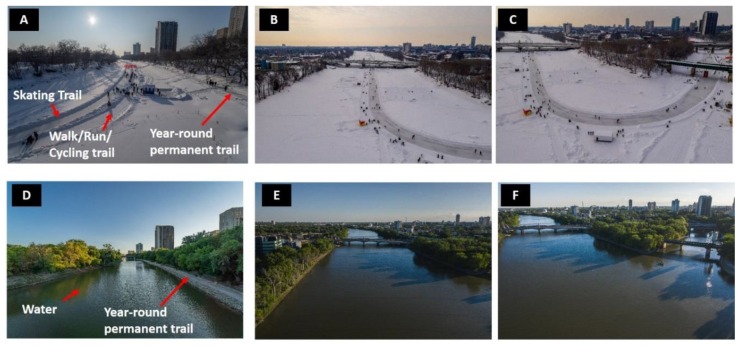
Visual evidence that the intervention consisted of walking (running, skating, and cycling) on water. (**A**–**C**): Representation of the frozen waterway at different segments (**A**) Assiniboine River Heading South; (**B**) Red River heading south; (**C**) Red and Assiniboine River Junction. (**D**–**F**): Representation of the same points along the waterway in July 2019 providing evidence that the trail network was created on water.

**Figure 2 ijerph-16-03627-f002:**
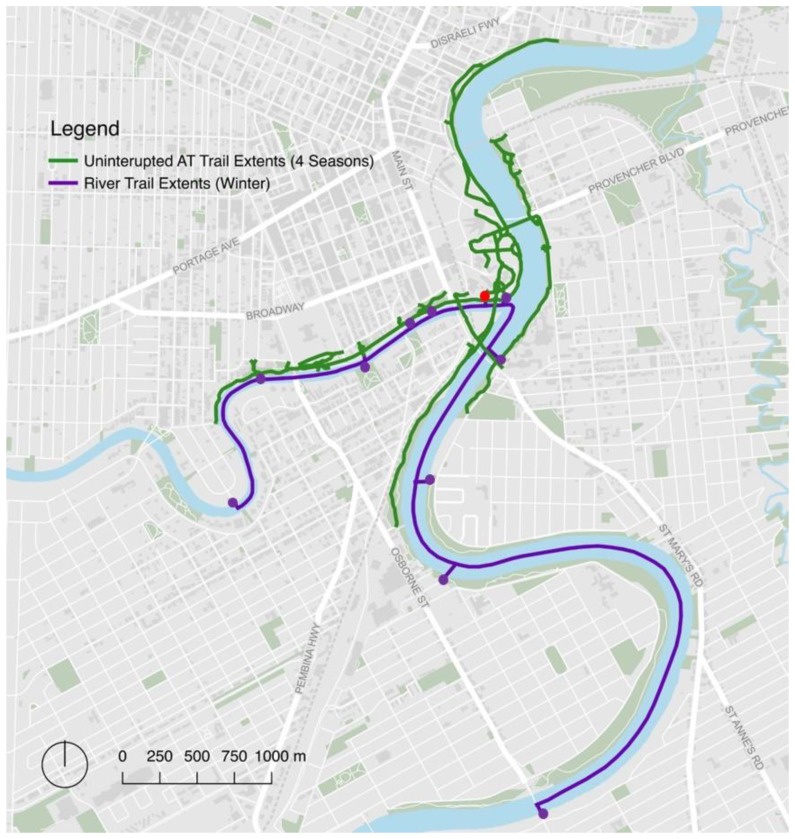
Map of trail network. Red dot = location of PYRO-Box Eco-Counter; purple dots = access points for the frozen waterway trail; green lines = permanent trail network accessible for 12 months of the year; purple line = frozen waterway trail.

**Figure 3 ijerph-16-03627-f003:**
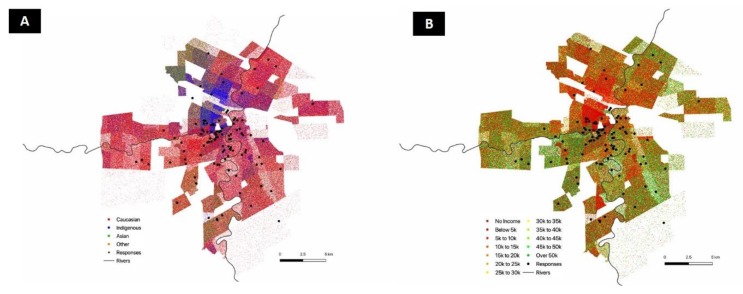
Geomap of trail users. Neighbourhoods from which trail users originated mapped according to **A**—the geographic ethnic make-up of the city of Winnipeg and **B**—the geographic household income of the city of Winnipeg.

**Figure 4 ijerph-16-03627-f004:**
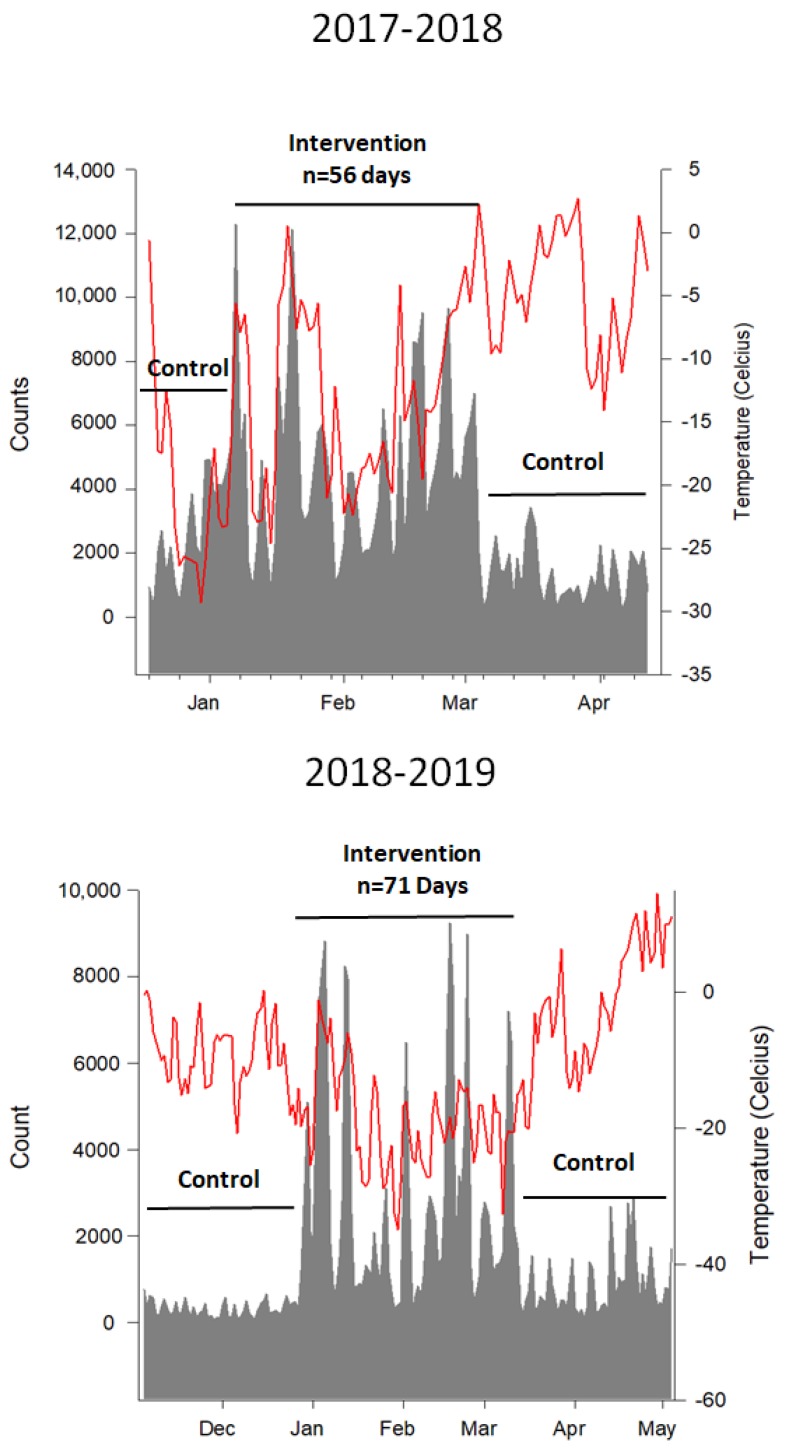
Objectively measured trail visits before, during, and after the opening of the frozen waterway trail. Red line = temperature (Y2 axis); grey filled line = daily user counts during control and intervention periods. Lines for control and intervention periods correspond to dates of data collection.

**Figure 5 ijerph-16-03627-f005:**
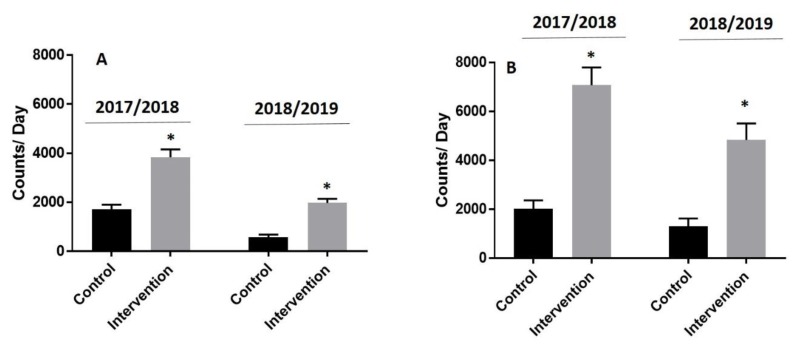
Mean daily trail visits during intervention and control periods stratified by type of day. A = weekday day; B = weekend day; * = *p* < 0.001.

**Figure 6 ijerph-16-03627-f006:**
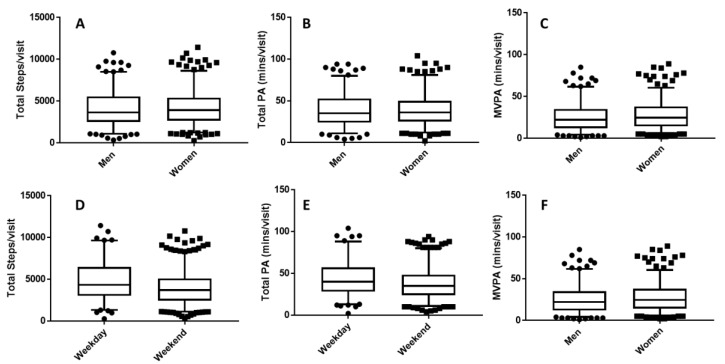
Objectively measured physical activity during a single visit to the frozen waterway trail. Data presented are medians with 95% confidence intervals. Additional dots are outliers. A/D—total steps; B/E—total minutes of activity; C/F—total moderate to vigorous intensity physical activity (MVPA).

**Table 1 ijerph-16-03627-t001:** Natural experiment details. CI, confidence interval.

Variable	2017/2018			2018/2019		
	Pre-Control	Intervention	Post-Control	Pre-Control	Intervention	Post-Control
Start Date	08/12/2017	07/01/2018	04/04/2018	03/12/2018	04/01/2019	11/03/2019
End Date	06/01/2018	03/03/2018	04/04/2018	03/01/2019	10/03/2019	12/04/2019
Total Days	20	56	40	31	67	33
Weekend Days	5	16	11	8	20	8
Mean Temp % (95% CI)	−20 °C (−23 °C,−16 °C)	−13 °C (−15 °C,−11 °C)	−4 °C (−6 °C,−3 °C)	−10 °C (−13 °C,−8 °C)	−17 °C (−18 °C,−15 °C)	−1 °C (−2 °C,+1 °C)
User Counts	51,183	266,581	41,728	25,849	182,298	21,709

**Table 2 ijerph-16-03627-t002:** Trail user demographics.

Variable	Females (*n* = 127)	Males (*n* = 86)	Total (%) (*n* = 218)
Household Income			
<$15,000	9	5	14 (6.4%)
$15,000–49,000	23	13	36 (16.5%)
$50,000–74,999	18	13	31 (14.2%)
$75,000–99,999	16	16	32 (14.7%)
>$100,000	39	29	68 (31.2%)
Prefer not to disclose			37 (17.0%)
Ethnicity			
Caucasian	103	61	164 (75.2%)
Indigenous	9	7	16 (7.4%)
Asian	7	10	17 (7.8%)
Other	7	4	11 (5.0%)
Prefer not to disclose			10 (4.6%)
Age (years)			
18–24	8	4	12 (5.5%)
25–34	35	20	55 (25.2%)
35–44	32	18	50 (23.0%)
45–64	44	33	77 (35.3%)
>65	5	8	13 (6.0%)
Prefer not to disclose			11 (5.0%)

**Table 3 ijerph-16-03627-t003:** Self-reported information on trail use.

Variable	Females (*n* = 127)	Males (*n* = 86)	Total (%) * (*n* = 218)
Reason for use			
Transportation	1	3	4 (1.8%)
Exercise/Recreation	102	60	164 (75.2%)
Both	24	23	50 (23.0%)
Timing of first visit			
<3 months ago	27	15	42 (19.3%)
4–11 months ago	2	1	3 (1.3%)
1–3 years ago	17	13	30 (13.8%)
>3 years ago	81	57	143 (66.6%)
Travel time to trail			
<5 min	22	11	34 (15.6%)
6–15 min	56	41	100 (45.9%)
16–29 min	35	25	61 (28%)
>30 min	14	9	23 (10.5%)
Average duration per visit			
<30 min	1	9	10 (4.6%)
30–44 min	28	18	47 (21.6%)
45–59 min	46	14	62 (28.4%)
1–2 h	48	42	91 (41.7%)
>2 h	4	3	8 (3.7%)
No. visits in the past			
1 day	74	42	119 (54.6%)
2–3 days	45	28	74 (33.9%)
4–5 days	4	14	19 (8.7%)
6–7 days	4	2	6 (2.8%)

* Note: five users preferred not to disclose their gender and data were removed from the table.

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
