# Peer review of "Walking on Water—A Natural Experiment of a Population Health Intervention to Promote Physical Activity after the Winter Holidays"

_ijerph, 2019, doi:10.3390/ijerph16193627_

Round 1
Reviewer 1 Report
Walking on Water A natural experiment of a population health intervention to promote physical activity after the winter holidays
The study aims to assess the impact of an ice trail on user visits and estimate physical activity levels and sociodemographic data of the ice trail users.
Introduction
The research questions have been developed on the assumption that the winter holiday leads to excessive weight gain in the population. This is both conceivable and well documented in the Introduction.
Design and Methods
The methods used seems suitable. The pre and post recordings of users are well placed in time and with a suitable duration to serve as a control for the intervention data. The placement of the PYRO-Box counter should have been clearly marked on the map (Figure 2), as well as the access points (a more thorough explaining text would help). The statistical analysis is advanced, but does not always show up in the results. The use of AIC and Bayesian statistics could have been more presented. The Ljugn-Box test is probably unknown to many readers (at least to me), so it deserves some explanation. IBM SPSS is mentioned by version number, so also should R be. By the way, address for SPSS is since version 19 Arnmonk, NY, USA (http://www-01.ibm.com/support/docview.wss?uid=swg21476197). The use of pedometers should be more explained, especially as also skating was an activity of choice, how well do the pedometer record the gliding movement of skating?
Results
Table 1 presents important overview data. The date format is MMDDYY, I believe the journal uses DDMMYY, or the month could be spelled out to avoid confusion. Figure 3 is difficult to read, due to both size and grey-tones. Here colours would be welcomed. Figure 4 and 5 seems to me to nearly double each other, and I find Figure 5 to be more easily readable. Again, colour and size could enhance Figure 4 also. If Figure 4 is kept, a more informative legend should be applied, clearly identifying temperature data and visits numbers. The placement of the lines underscoring Control and Intervention seems rather arbitrary, or is it a system there based on numbers of users? If so, the legend should point this out. The Figure 5 legend is also missing important information, is panel A weekdays and panel B weekends?
Discussion
The “meaningful dose” of activity achieved by users should be discussed more, since pedometer recordings can be questionable when estimating energy expenditure in a variety of activity forms.
In the Discussion, estimated energy expenditure is presented. This should also have been presented in the Results and in Design and Methods: which formulas?). Also there is reference to “the large effect size” (line 249), but I cannot find it as a result. The earliest recordings of skating may have been 4.000 years ago (https://en.wikipedia.org/wiki/Ice_skating) (Line 212). Check line 236, last words, is that correct?
In all, this is a nice study that present important new knowledge on how to be physical active.
Author Response
Thank you for the rapid and insightful review of our manuscript. The suggested revisions have significantly strengthened the manuscript. Please accept the following detailed response to reviews.
Comment #1: "The placement of the PYRO-Box counter should have been clearly marked on the map (Figure 2), as well as the access points (a more thorough explaining text would help)."
Response: We have added information for the location of the Pyro-box and the entry points for the waterway trail on the map in Figure 2 and in the methods (lines 91,92, 97,99 and 115)
Comment #2: "The statistical analysis is advanced, but does not always show up in the results. The use of AIC and Bayesian statistics could have been more presented. The Ljugn-Box test is probably unknown to many readers (at least to me), so it deserves some explanation."
Response: Revised accordingly. Information for AIC and Ljugn-Box test were added to the results section to describe the goodness of fit of the models and describe the randomness of autocorrelation in the time series. (line 152 and line 176-178)
Comment #3: "IBM SPSS is mentioned by version number, so also should R be. By the way, address for SPSS is since version 19 Arnmonk, NY, USA (http://www-01.ibm.com/support/docview.wss?uid=swg21476197)."
Response: Revised accordingly (line 153-154)
Comment #4: "The use of pedometers should be more explained, especially as also skating was an activity of choice, how well do the pedometer record the gliding movement of skating?"
Response: We have provided additional details on the use pedometers and that their use for skating has not be validated. We have highlighted this as a limitation in the discussion section (line 280).
Comment (s) #5:
A: The date format is MMDDYY, I believe the journal uses DDMMYY, or the month could be spelled out to avoid confusion.
Response: Revised Accordingly
B- "Figure 3 is difficult to read, due to both size and grey-tones. Here colours would be welcomed. "
Response: Revised accordingly. Figure is colour coded. Could be expanded into individual figures if editorial staff thought it would enhance readibility.
C- "Figure 4 and 5 seems to me to nearly double each other, and I find Figure 5 to be more easily readable. Again, colour and size could enhance Figure 4 also. If Figure 4 is kept, a more informative legend should be applied, clearly identifying temperature data and visits numbers. "
Response : We would prefer to keep Figure 4. We can enlarge the figures to improve readibility and have provided a legend to facilitate interpretation. (Line 182)
D- "The placement of the lines underscoring Control and Intervention seems rather arbitrary, or is it a system there based on numbers of users? If so, the legend should point this out. The Figure 5 legend is also missing important information, is panel A weekdays and panel B weekends?"
Response: Legends have been added to the figures
Comment #6: "The “meaningful dose” of activity achieved by users should be discussed more, since pedometer recordings can be questionable when estimating energy expenditure in a variety of activity forms."
Response: We have revised the discussion to temper the impact of the "dose" of physical activity on health outcomes (lines 246-260).
Comment #7: In the Discussion, estimated energy expenditure is presented. This should also have been presented in the Results and in Design and Methods: which formulas?). Also there is reference to “the large effect size” (line 249), but I cannot find it as a result. The earliest recordings of skating may have been 4.000 years ago (https://en.wikipedia.org/wiki/Ice_skating) (Line 212). Check line 236, last words, is that correct?
Response: We have provided a reference to the manuscript for the formula used to estimate energy expenditure. The reference to the large effect size was removed and replaced with a statement about the use of objective measures (Line 272) We have revised the first line of paragraph 2 to read "recreational physical activity". Line 236 is correct. See Table 3: Journal of Physical Activity and Health, 2011, 8, 1004 -1013.
Reviewer 2 Report
I would like to commend the authors on the completion of such a novel research project, with a significant sample size. The information provide makes for interesting reading and demonstrates the benefits of a different approach to physical activity. The manuscript is well written and contains some thought-provoking analysis. With that said I do have a few questions that need to be considered as a means of improving the manuscript. Of primary concern is that no attempt appears to be made to ascertain the reason for the participants visit. Was any data collected as to why they were undertaking the trail? Did these people use the trail because it was frozen or because they had more time available (i.e. on winter vacation or leave from work)? Similarly was any data collected regarding enjoyment during the trail usage? This will impact their potential desire to return and as such allow for more concrete evidence that this is an appropriate means of promoting physical activity.
There are also a number of substantial claims made in the discussions that are not directly evident from the results. Specifically now assessment as to why there has been a 3-5 fold increase in visitors during this period (L.214).
L.216-218 - this is a significant leap in assumption. Can the authors confirm how there findings demonstrate these to be "the most impactful" strategy?
L.223-224 - Can the authors provide a reason for this?
L.238-239 - What is the time frame require to undertake 17-43 visits per person (i.e. in a month, two months)? Is this feasible?
Author Response
Thank you for the rapid and insightful review of our manuscript. The suggested revisions have significantly strengthened the manuscript. Please accept the following detailed response to reviews.
Comment #1: "Of primary concern is that no attempt appears to be made to ascertain the reason for the participants visit. Was any data collected as to why they were undertaking the trail? " Did these people use the trail because it was frozen or because they had more time available (i.e. on winter vacation or leave from work)?
Response: Data for rationale for trail use is provided in Table 3. 75% of users were accessing the trail for recreation and/exercise. Most people accessed the trail on the weekend (Figure 5) when they were off from work.
Comment #2: Similarly was any data collected regarding enjoyment during the trail usage?
Response: We did not ask about enjoyment, however in section 3.3 of the results the majority felt the trail improved their mental/emotional well-being.
Comment #3: There are also a number of substantial claims made in the discussions that are not directly evident from the results. Specifically now assessment as to why there has been a 3-5 fold increase in visitors during this period (L.214).
Response: We have presented the data for the 3-5-fold increase in trail use in Figures 4 and 5, demonstrating large increases in daily trail use. We believe the increase was due to the fact that the frozen waterway is a novel and attractive trail.
Comment #4: L.216-218 - this is a significant leap in assumption. Can the authors confirm how there findings demonstrate these to be "the most impactful" strategy?
Response: We have removed the terms "most impactful". and revised the sentence to read "The data presented here suggest that creating a trail along a frozen waterway is a a unique population health strategies that attracts large segments of an urban population to be physically active in the winter." (Line 234-236)
Comment #5: L.223-224 - Can the authors provide a reason for this?
Response: We cannot explain why population health interventions tend to favor more affluent segments of the population, however it may be a function of political will, underrepresentation of underserved populations in the decision making process or features of poor urban areas.
Comment #6 What is the time frame require to undertake 17-43 visits per person (i.e. in a month, two months)? Is this feasible?
Response: Assuming a person visits 3-5 times per week, they could achieve this within 5-10 weeks. We have added a comment to the discussion. (line 262)

Reviewer 3 Report
It’s a very important topic to examine the effectiveness of frozen waterway to promote physical activity in the winter season. The authors did a good job on addressing the importance of this health intervention, as well as presenting the results of the quasi-experiment. However, I think there are some areas that need further discussion to improve the quality of the manuscript.
1. I suggest the authors to list the hypotheses and/or research questions of the study explicitly in separate paragraphs, instead of embedding them with literature review paragraphs. For example:
H1: XXXXXXXXXXXXXXXXXX.
RQ1: XXXXXXXXXXXXXXXXXX?
RQ2: XXXXXXXXXXXXXXXXX?
The authors referred to these hypotheses or research questions multiple times in the following sections, yet it’s hard to identify which H or RQ were discussed.
2. In the study, the authors used three main variables to examine the effectiveness of the intervention, which are the user visits, their physical activity, and their perceived benefits. More literature is needed for the reason of the adoption of these variables. Why these variables could be used to indicate the success of a PA intervention? Are they being used as measurements widely in PA interventions? Are there other measurements for the effectiveness of PA interventions? If so, why others were not being selected?
3. Based on the data presented, it seems the user visits dropped a lot in the 18/19 period compared to the 17/18 period, no matter during the intervention times or the control times. Did the authors conduct any analysis to compare the yearly difference? Is it significant? What are the potential reasons for this drop? Does it indicate a potential sustainability issue with visitors for this winter waterway design?
4. In the discussion section, the authors talked about the potential equity problem that less minorities or users with low income were observed in the study. Is it because of the location of this waterway trial? Is it close to low income communities? I would like to see more discussion about the authors’ thoughts on the potential solutions for the inclusion for equity.
Author Response
Comment #1: I suggest the authors to list the hypotheses and/or research questions of the study explicitly in separate paragraphs, instead of embedding them with literature review paragraphs. For example: H1: XXXXXXXXXXXXXXXXXX. RQ1: XXXXXXXXXXXXXXXXXX? RQ2: XXXXXXXXXXXXXXXX? The authors referred to these hypotheses or research questions multiple times in the following sections, yet it’s hard to identify which H or RQ were discussed.
Response: Revised accordingly. (line 63-68)
Comment #2: Why these variables could be used to indicate the success of a PA intervention? Are they being used as measurements widely in PA interventions? Are there other measurements for the effectiveness of PA interventions? If so, why others were not being selected?
Response: These are the most common methods to measuring population-based physical activity interventions, however they are rarely used in concert. The justification for using these three simultaneously is provided in the methods section (lines 131-135).
Comment #3: Based on the data presented, it seems the user visits dropped a lot in the 18/19 period compared to the 17/18 period, no matter during the intervention times or the control times. Did the authors conduct any analysis to compare the yearly difference? Is it significant? What are the potential reasons for this drop? Does it indicate a potential sustainability issue with visitors for this winter waterway design?
Response: We have added a description of the conditions that may explain this difference. We do not believe it will influence the sustainability of the intervention. Line 187-190: Counts in the 2018/2019 winter season were significantly lower than in the 2017/2018 season (266,581 vs 182298, p < 0.01). This was likely due to a colder winter season (mean daily temperature -170C vs -130C) and a prolonged climactic event known as a Polar Vortex during the months of January and February.
Comment #4: In the discussion section, the authors talked about the potential equity problem that less minorities or users with low income were observed in the study. Is it because of the location of this waterway trial? Is it close to low income communities? I would like to see more discussion about the authors’ thoughts on the potential solutions for the inclusion for equity
Response: We have added 2 lines in the discussion in an attempt to explain why this difference in socio-demographic participation occurred. (line 255-258) The reason for these differences is unclear, but may relate to difficulties accessing the trail network with public transit or due to lack of adequate clothing needed to participate in physical activity during cold days. It may also be that the elements of the frozen waterway were not designed in a way that made it attractive for these segments of the population.

Round 2
Reviewer 2 Report
The authors have addressed the major concerns identified in the initial review, resulting in an improved quality of the manuscript.
Author Response
Thank you for approving the revisions to the original submission.
Reviewer 3 Report
The authors addressed well for the major concerns of the reviewers, and I believe the manuscript has been improved from the edits.
However, I was not able to find the edits for Comment #2 from Reviewer 3 in the revised manuscript. Lines 131-135 in the revised manuscript was not about the justification of the measurements used in the study. I also did not find this information by checking the highlighted changes through the manuscript. I think the authors need to provide further clarification for this.
Author Response
Thank you for a second thorough review of the manuscript.
Comment #2: Why these variables could be used to indicate the success of a PA intervention? Are they being used as measurements widely in PA interventions? Are there other measurements for the effectiveness of PA interventions? If so, why others were not being selected?
In section 2.7 of the methods: "Outcomes of Interest", we state the following with supporting references:
"Trail counts, objective measurements of physical activity and surveys of users are the most common methods used to assess physical activity-based population health interventions. 16, 17,18,19,20, 21 When used in concert, data can be triangulated to answer questions, relevant to policy makers, including: How often is the intervention being used?; What dose/amount of activity are users achieving?; and What are users saying about the intervention?"
If additional information is required from the review to justify the use of the three different methods for assessing population-based physical activity interventions, we can certainly try to provide additional justification for the methods used.